# The Effect of Spring Barley Fertilization on the Content of Polycyclic Aromatic Hydrocarbons, Microbial Counts and Enzymatic Activity in Soil

**DOI:** 10.3390/ijerph20053796

**Published:** 2023-02-21

**Authors:** Ewa Mackiewicz-Walec, Sławomir Józef Krzebietke, Agata Borowik, Andrzej Klasa

**Affiliations:** 1Department of Agrotechnology and Agribusiness, Faculty of Agriculture and Forestry, University of Warmia and Mazury in Olsztyn, 10-719 Olsztyn, Poland; 2Department of Agricultural and Environmental Chemistry, Faculty of Agriculture and Forestry, University of Warmia and Mazury in Olsztyn, 10-719 Olsztyn, Poland; 3Department of Soil Science and Microbiology, Faculty of Agriculture and Forestry, University of Warmia and Mazury in Olsztyn, 10-727 Olsztyn, Poland

**Keywords:** manure, mineral fertilization, soil microbiome, soil enzymes, long-term controlled experiment, PAHs

## Abstract

Soil-dwelling microorganisms play an important role in the environment by decomposing organic matter, degrading toxic compounds and participating in the nutrient cycle. The microbiological properties of soil are determined mainly by the soil pH, granulometric composition, temperature and organic carbon content. In agricultural soils, these parameters are modified by agronomic operations, in particular fertilization. Soil enzymes participate in nutrient cycling and they are regarded as sensitive indicators of microbial activity and changes in the soil environment. The aim of the present study was to determine whether PAH content in soil is associated with the microbial activity and biochemical properties of soil during the growing season of spring barley treated with manure and mineral fertilizers. Soil samples for analysis were collected on four dates in 2015 from a long-term field experiment established in 1986 in Bałcyny near Ostróda (Poland). The total content of PAHs was lowest in August (194.8 µg kg^−1^) and highest in May (484.6 µg kg^−1^), whereas the concentrations of heavier weight PAHs was highest in September (158.3 µg kg^−1^). The study demonstrated that weather conditions and microbial activity induced considerable seasonal variations in PAHs content. Manure increased the content of organic carbon and total nitrogen, the abundance of organotrophic, ammonifying and nitrogen-fixing bacteria, actinobacteria and fungi and enhanced the activity of soil enzymes, including dehydrogenases, catalase, urease, acid phosphatase and alkaline phosphatase.

## 1. Introduction

Environmental pollution caused by polycyclic aromatic hydrocarbons (PAHs) poses one of the greatest problems in the contemporary world. Polycyclic aromatic hydrocarbons (PAHs) are considered to be especially toxic to humans, likewise to plants, microorganisms and other living organisms. PAH toxicity is a well-known fact, especially its ability to cause cancer [1,2,3,4,5]. Polycyclic aromatic hydrocarbons belong to the group of persistent organic pollutants. These highly toxic compounds are accumulated in soil and persist in the environment for long periods of time [6,7]. These compounds are generated during incomplete combustion of organic matter in natural and anthropogenic processes [8,9]. Polycyclic aromatic hydrocarbons are classified into two main groups based on their chemical structure: low-molecular-weight (LMW) PAHs that contain two or three aromatic rings and high-molecular-weight (HMW) PAHs that contain four or more aromatic rings. Low-molecular-weight PAHs are relatively easily degraded, whereas most HMW PAHs with fused rings are carcinogenic and much more difficult to decompose [7].

The microbial degradation of PAHs is influenced by various environmental factors, including the availability of nutrients, the abundance and type of soil-dwelling microorganisms, as well as the type and chemical properties of degraded PAHs. Polycyclic aromatic hydrocarbons can be potentially degraded/transformed by a wide range of bacterial and fungal species [10,11]. Microorganisms easily adapt to new environmental conditions and they derive energy and nutrients from compounds that are not products of their own metabolism. This observation implies that microorganisms could be effectively used to reduce PAH levels in soil. Microorganisms that are potentially useful in soil remediation can be divided: autotrophs that derive carbon from carbon dioxide and heterotrophs that obtain carbon from the degradation of organic matter from both natural and anthropogenic sources [12]. Polycyclic aromatic hydrocarbons are sources of carbon and energy for microorganisms [3] and their content in soil can be effectively reduced through the addition of organic matter which that microbial activity [13,14].

Low nutrient availability can also decrease the effectiveness of bioremediation in areas contaminated with PAHs. In addition to easily metabolized sources of carbon, microorganisms also require minerals, including nitrogen, phosphorus, potassium and iron, for metabolic and growth processes. Therefore, contaminated and nutrient-deficient soils should be supplemented to stimulate the growth of autochthonous microorganisms [7,10]. According to Ravanipour [15], nutrient application can be regarded as the most important factor in bioremediation strategies for removing PAHs from soil.

Dissolved organic matter (DOM) is a major source of organic carbon (Corg) in soil and it plays a key role in carbon cycling. The presence of strong bonds between PAHs and soil organic matter (SOM) can significantly decrease the bioavailability and mobility of PAHs. As a result, these pollutants tend to accumulate in carbon-rich organic soils rather than in the deeper strata of mineral soils [16,17].

Organic matter increases soil moisture content and stimulates microbial growth. At the same time, organic and mineral nutrients enhance the abundance of exogenous microorganisms in the soil microflora, which increases the counts and viability of bacteria and other organisms capable of degrading PAHs [18].

There are several biological remediation techniques (bioremediation; bacteria and fungi, phycoremediation; algae, phytoremediation; plants and rhizoremediation; plant and microbe) for the treatment of PAH-contaminated soil. Based on the selection of the proper remediation approach, these remediation techniques are carried out by two basic types: (i) in situ (land farming, biostimulation, bioaugmentation, composting and phytoremediation) and (ii) ex situ (bioreactors) [19].

The rate of PAH biodegradation is affected by pH, which affects the development of soil microorganisms and enzymes. An increase in soil acidity promotes the accumulation of PAHs in soil [11,20]. The persistence of PAHs containing three and four aromatic rings increases in acidic soils. Liming can slow down PAH decomposition, depending on soil parameters, environmental factors and the properties of PAHs [21,22]. Most microorganisms are sensitive to pH and have a preference for pH-neutral environments (6.5–7.5) [10]. Neutralization of soil pH increases bacterial abundance and promotes the decomposition of PAHs [23].

Environmental contamination with persistent organic pollutants has emerged as a serious threat of pollution. Scientific knowledge upon microbial interactions with individual pollutants over the past decades has helped to abate environmental pollution [24]. For the last four decades, the degradation of PAHs by microorganisms has been well studied; most of the reported work has been focused primarily on the biodegradation of PAHs containing two to four fused rings [25]. Limited work has been dedicated to HMW PAHs. The influencing mechanism of soil fertilization PAH biodegradation is still unclear, especially microbe counts and soil enzyme activities.

In the natural environment, organic compounds are degraded by soil microorganisms and enzymes both under aerobic and anaerobic conditions [9]. Intermediate decomposition products are often more toxic for microorganisms, animals and humans than the parent compounds. The presence and accumulation of PAHs in soil has not been extensively studied to date and further research is needed to address this problem. Therefore, the aim of this study was to evaluate the influence of long-term varied organic mineral and mineral fertilization during the growing season and after the harvest of spring barley grown in the eighth cycle of crop rotation, on the microbial activity and biochemical properties of soil and the accumulation of PAHs in soil.

The research was conducted in order to assess the effect of long-term fertilization with manure and mineral fertilizers on the content of polycyclic aromatic hydrocarbons (PAHs) in soil. Relationships were also explored between the soil content of PAHs and the soil microbiological (counts of bacteria and fungi) and biological activity (enzymatic activity). The combined application of manure and mineral fertilizers has been studied in only a very few research experiments, hence their effect on PAH content in soil is still unexplored. The new insights contribute to a better understanding of PAH biodegradation processes under complex natural conditions. It was assumed that the optimal fertilization both with manure and with mineral fertilizers customized strictly to nutritional requirements to field crops does not exceed the permissible concentrations of the assessed PAHs in the soil.

## 2. Materials and Methods

### 2.1. Research Location and Experimental Design

Soil samples for the study were obtained in 2015 from a long-term controlled field experiment established in Bałcyny, Poland (N: 53°35′38.1″, E: 19°50′56.1″) in 1986. The experiment was conducted in three replicates (blocks) on soil developed from sandy loam (Haplic Luvisols, IUSS Working Group [26]), according to a previously described design [27]. The soil nutrition regime included the application of manure and mineral fertilizers or mineral fertilizers only. The same amount of nutrients was supplied with mineral fertilizers in both systems. The following mineral fertilizers were applied in the production of spring barley (*Hordeum vulgare* L.): (1) N_0_P_0_K_0_, (2) N_1_P_1_K_1_, (3) N_2_P_1_K_1_, (4) N_3_P_1_K_1_, (5) N_2_P_1_K_2_, (6) N_2_P_1_K_3_, (7) N_2_P_1_K_2_Mg, (8) N_2_P_1_K_2_MgCa (N_1_-30, N_2_-60, N_3_-90, P_1_-34.9, K_1_-33.2, K_2_-66.4, K_3_-99.7, Mg-18.1 kg ha^−1^) (Table 1).

The following crops were grown in rotation: sugar beets, spring barley, maize and spring wheat. After the spring wheat harvest, soil was limed with 2.5 t CaO ha^−1^ two years before the spring barley cultivation. Before the study, soil composition (per kg) was as follows: 100.0 mg of K, 53.2 mg of Mg, 41.3 mg of P, 7.9 g of organic carbon and 0.79 g of total nitrogen. Soil pH was slightly acidic (6.2 in 1 mol dm^−3^ KCl). Spring barley was grown in the second year after manure application (at the rate of 40 t ha^−1^). The content of nutrients, heavy metals and PAHs (LMW and HMW) in the manure was described previously by Krzebietke et al. [3]. All samples were analyzed for the 16 PAH priority pollutant listed by US EPA [28].

Soil samples for analyses of chemical, biochemical and microbiological properties and PAH levels were collected at a depth of 0–30 cm on four dates during the growing season of spring barley (BBCH-10, BBCH-23), after harvest and after skimming. Fresh soil samples for microbiological and biochemical analyses were passed through a sieve with a 2 mm mesh size directly after they had been transported to the laboratory.

In a study by Smreczak and Maliszewska-Kordybach [29], spring barley was most susceptible to soil contamination with selected PAHs in comparison with other crop species (maize, white mustard, sunflower). Therefore, soil samples for the analyses of microbiological and biochemical parameters and PAH content were collected in 2015 when spring barley was grown in rotation. Agronomic practices were applied in accordance with the requirements of the tested crop (Appendix A). Phenological observations were conducted during the growing season of spring barley and the main developmental stages are described in Appendix A.

### 2.2. Chemical Analyses of Soil

Selected chemical properties of soil (pH, Hh, total N, Corg) were analyzed. The following parameters were determined in air-dried soil samples: pH 1 mol KCl∙dm^−3^, by the potentiometric method; hydrolytic acidity (Hh), by Kappen’s method; total nitrogen content, by distillation after mineralization in sulfuric (VI) acid with the addition of the selenium reagent mixture; organic carbon content, by the Kurmies method.

The content of 16 PAHs was determined with the Trace GC/MS Ultra ITQ900 system with a TRIPlus autosampler (Thermo Fisher Scientific, Waltham, MA, USA) and a flame ionization detector. The total content of 16 PAHs (naphthalene, acenaphthene, acenaphthylene, fluorene, phenanthrene, anthracene, fluoranthene, pyrene, benzo(a)anthracene, chrysene, benzo(b)fluoranthene, benzo(k)fluoranthene, benzo(a)pyrene, indeno(1,2,3-cd)pyrene, dibenzo(a,h)anthracene and benzo(g,h,i)perylene) was determined by the method described by Krzebietke et al. [3]. The content of LMW PAHs (naphthalene, acenaphthene, acenaphthylene, fluorene, anthracene, phenanthrene, fluoranthene, pyrene and chrysene) and HMW PAHs (benzo(a)anthracene, benzo(a)pyrene, benzo(b)fluoranthene, benzo(k)fluoranthene, benzo(g,h,i)perylene, indeno(1,2,3-cd)pyrene and dibenzo(a,h)anthracene) was determined in soil samples.

### 2.3. Microbiological and Biochemical Analyses of Soil

The counts of the following soil-dwelling microorganisms were determined in the soil samples: organotrophic bacteria, on Bunt and Rovira agar [30]; ammonifying and nitrogen-fixing bacteria, on the medium described by Wyszkowska [31]; actinobacteria, on the medium described by Küster and Williams with the addition of nystatin and actidione [32]; fungi, on Martin’s agar [33].

Microbial counts were determined by plating in three replicates. Microbial cultures were incubated at a temperature of 28 °C. The number of colony-forming units (CFU) was determined with a colony counter. The activity of the following soil enzymes was determined in three replicates: dehydrogenases, by the method described by Öhlinger [34]; urease, acid phosphatase and alkaline phosphatase, by the method described by Alef and Nannipieri [35]; catalase, by the method described by Johnson and Temple [36]. Microorganisms were isolated with the serial dilution method following the procedure described in the study by Wyszkowska et al. [37]. The procedure for the determination of soil enzymatic activity was presented in the study by Borowik et al. [38] and microbial counts. The culture conditions and the exact procedure for the isolation of microorganisms were described in our earlier paper in the study by Borowik et al. [39].

### 2.4. Statistical Analysis

The data (content of LMW PAHs and HMW PAHs and total content of 16 PAHs) were processed statistically by repeated measures ANOVA, where manure application and varied mineral fertilization were the fixed grouping factors and the sampling date was the repeated measure factor:(1)$$y_{ijkl}=\mu +\tau_{i}+f_{k}+{\left(\tau f\right)}_{ik}+Date_{\;l}+{\left(\tau Date\right)}_{il}+{\left(fDate\right)}_{kl}+{\left(\tau fDate\right)}_{ikl}+\beta_{j}+{\left(\beta Date\right)}_{jl}+\varepsilon_{ijkl}$$
where μ is the general average; $\tau_{i}$ is the effect of manure and NPK *i*; $f_{k}$ is the effect of manure application *k;* $\beta_{j}$ is the blocking effect *j; Date_l_* is the repeated measures effect; ${\left(\tau f\right)}_{ik}$ is the effect of the interactions between the *i*th rate of NPK fertilization and manure *k*; ${\left(\tau Date\right)}_{il}$ is the effect of the interactions between the *i*th rate of NPK fertilization and sampling date *l*; ${\left(fDate\right)}_{kl}$ is the effect of the interactions between manure *k* and sampling date *l*; ${\left(\beta Date\right)}_{jl}$ is the effect of the interactions between blocks and sampling date *l*; ${\left(\tau fDate\right)}_{ikl}$ is the effect of the interactions between the *i*th rate of NPK fertilization, manure *k* and sampling date *l*; $\varepsilon_{ijkl}$ is the random error with normal distribution, expected value 0 and variance σ2.

Before performing statistical analyses, dependent variables in each group were tested for normal distribution. The homogeneity of variance was determined in groups and the sphericity (equality of variances of the differences between measurements) was evaluated with Mauchly’s test. Data that did not satisfy the sphericity condition were analyzed with the use of Wilk’s lambda test and Pillai’s trace criterion.

The Shapiro–Wilk test revealed that the data did not have normal distribution, therefore they were log transformed. In the next step, data were processed in Tukey’s post hoc HSD test at *p* < 0.05.

Microbial counts and enzymatic activity were evaluated with the Kruskal–Wallis non-parametric test for independent samples. The analyses were performed on untransformed data. The relationships between soil microbial activity, biochemical properties, organic carbon and total nitrogen content vs. PAH content (17 parameters) were determined by principal component analysis (PCA). The strength of the correlations in PCA was validated with the use of Bartlett’s test of sphericity. The number of principal components was selected with the Kaiser criterion based on eigenvalues greater than one (λi > 1). The interpretation of individual principal component PCi was simplified by varimax rotation. The results of all chemical, microbiological and biochemical analyses were interpreted by focusing on the main effects. All statistical analyses were performed in the Statistica 13 program [40].

### 2.5. Weather Conditions

Considerable variations in temperature and precipitation were noted in 2015 (Appendix A). Microorganisms require supportive weather conditions, including temperature and soil moisture content, which determine the rate of microbial growth and enzymatic activity of soil. Changes in temperature and precipitation were monitored for 7 days before each sampling date. Soil samples were collected for microbiological and biochemical analyses on four dates (22 April, 18 May, 8 August and 15 September 2015) and weather conditions varied considerably in each monitoring period. The lowest temperature (6.2 °C) was observed before the first sampling date, whereas the temperature before the second sampling date was 1.9 times higher. In contrast, precipitation during the 7 days preceding the sample collection was 1.8 times lower in May than in April. The least favorable weather conditions were noted in August (7 days before sampling), which was characterized by a very high temperature (19.7 °C) and an absence of rainfall. In September, the temperature (12.6 °C) was only marginally higher than in April and precipitation levels (7.8 mm) were higher than in the remaining sampling periods.

## 3. Results and Discussion

### 3.1. Selected Chemical Parameters of Soil

#### 3.1.1. Soil pH and Hydrolytic Acidity

The growth of soil-dwelling microorganisms is determined by environmental conditions, including soil pH, which influences the microbiological and biochemical parameters of soil. Soil regularly amended with manure was characterized by higher pH values (in 1 mol KCl dm^−3^) and lower hydrolytic acidity than soil supplied with mineral fertilizers only (Table 1). Increasing nitrogen rates clearly decreased soil pH and increased hydrolytic acidity and the greatest changes in these parameters were observed under the influence of the highest nitrogen rate. In a study by Lemanowicz [41], high nitrogen rates and the absence of liming also undesirably increased hydrolytic acidity in soil. As expected, regular liming considerably increased soil pH and reduced hydrolytic acidity. The effects of liming were more pronounced in soil amended with manure than in soil supplied with mineral fertilizers only.

#### 3.1.2. Organic Carbon and Total Nitrogen Content

Carbon and nitrogen are essential for PAH degradation. Microorganisms have different nutritional requirements and various values of C:N ratios have been reported as optimal in the literature. Fungi dominate in soils with high C content and limited N supply. In turn, bacterial growth is influenced by both C and N content [42]. According to Amezcua-Allieri et al. [43], the C:N ratio affects the rate at which PAHs are removed from soil. Farahani et al. [44] reported that the rate of PAH degradation in soil is determined by the C:N ratio in the growth medium and the chemical form of nitrogen.

Organic carbon and total nitrogen are important indicators of soil fertility [45,46]. In the present study, the content of organic carbon and total N in soil was significantly influenced by manure and mineral fertilization (Table 2, Figure 1). Manure (M) exerted a significant effect and varied mineral fertilization (Min) and the interaction between these factors (M × Min) exerted highly significant effects on the total nitrogen content of soil. Nitrogen fertilization clearly increased the total nitrogen content of soil relative to the control treatment and enhanced the accumulation of Corg in soil in 2015 (Figure 2). The increase in soil Corg content in response to rising N rates can be attributed to the accumulation of biomass in soil after the harvest of each crop grown in rotation. Siwik-Ziomek and Lemanowicz [47] also reported an increase in the total nitrogen content of soil in response to increasing rates of N fertilizer. The value of the Block (B) parameter was not significant, which indicates that soil variability in the experimental field had no effect on the content of Corg and total N.

### 3.2. Microbiological and Biochemical Properties of Soil

#### 3.2.1. Microbial Abundance

##### Organotrophic Bacteria

In soil sown with spring barley, the counts of organotrophic bacteria were 1.7 higher in treatments that were regularly amended with manure than in treatments that were supplied with mineral fertilizers only (Figure 3a). The abundance of organotrophic bacteria increased with a rise in the N rate (Figure 3b). The highest N rate induced the greatest (1.4-fold) increase in the counts of organotrophic bacteria relative to the control treatment. The growth of organotrophic bacteria was also stimulated by higher potassium rates (66.4 and 99.7 kg∙ha^−1^). Liming decreased soil acidity and increased the availability of nutrients for organotrophic bacteria. Higher N and K rates induced similar effects. In 2015, the abundance of organotrophic bacteria in soil varied widely from 18,108 CFU kg^−1^ DM to 283,108 CFU kg^−1^ DM soil (Figure 3c). Bacterial counts were highest in soil samples collected in May (144,108 CFU kg^−1^ DM soil) and lowest in August (2.5 times lower). In April, the average abundance of organotrophic bacteria reached 69,108 CFU kg^−1^ DM soil and it was 19% lower than in September. May and September were characterized by the most favorable temperatures for bacterial growth (12.0 °C and 12.6 °C, respectively, during the 7-day monitoring period before sampling), which could explain the increase in the abundance of organotrophic bacteria in these months. According to Borowik et al. [48], organotrophic bacteria proliferate most rapidly at a temperature of around 15 °C.

##### Ammonifying Bacteria

Manure and mineral fertilizers significantly modified the abundance of ammonifying bacteria in soil (Figure 4a,b). The growth of these microorganisms was enhanced in treatments regularly amended with manure.

Potassium exerted varied effects on the counts of ammonifying bacteria; a moderate K rate decreased their abundance, whereas the highest K rate stimulated the proliferation of ammonifying bacteria (Figure 4b). Magnesium supplied with N_2_P_1_K_2_ had a minor influence on the counts of ammonifying bacteria. As expected, regular liming created the most favorable environment for the growth of ammonifying bacteria.

The abundance of ammonifying bacteria varied during the growing season (Figure 4c) and it was highest in August (139,108 CFU kg^−1^ DM soil) which was characterized by highly unfavorable weather conditions during the 7-day monitoring period before sampling (drought and very high temperature, 19.7 °C). According to Dąbek-Szreniawska et al. [49], a decrease in soil moisture content stimulates the growth of ammonifying bacteria. In the present study, the counts of ammonifying bacteria were 8% lower in May than in August and precipitation levels (3.9 mm) during the 7-day monitoring period before sampling were lower than in April and September (Appendix A). The abundance of ammonifying bacteria was lowest in April (65 × 10^8^ CFU kg^−1^ DM soil) and it was 15% higher in September (precipitation during the 7-day monitoring period before sampling reached 7.2 and 7.8 mm, respectively). The analyzed parameter was highest in May and August and it was considerably lower in April and September. These results could be attributed to optimal temperatures for microbial growth in May and August. Despite low precipitation in these months, soil water content was probably sufficient to promote the growth of ammonifying bacteria. Manure application increased the counts of ammonifying bacteria 1.4-fold relative to treatments supplied with mineral fertilizers only. The decomposition of organic matter supplied to soil with manure increased the content of mineral N and created a favorable environment for the development of soil microorganisms, including N-fixing bacteria. An increase in the content of mineral N as well as higher microbial counts promoted N immobilization in soil.

##### Nitrogen-Fixing Bacteria

Manure significantly increased the counts of N-fixing bacteria in soil (Figure 5a). The abundance of N-fixing bacteria was 1.4-fold higher in soil amended with manure every other year than in soil supplied with mineral fertilizers only. The decomposition of organic matter supplied with manure increased the content of mineral N. It should also be noted that manure creates favorable conditions for the growth of soil-dwelling microorganisms, including N-fixing bacteria. An increase in the content of mineral N and higher microbial counts promoted N immobilization in soil.

Increasing N rates exerted a minor effect on the abundance of N-fixing bacteria in soil (Figure 5b). Potassium was a more influential factor and higher K rates stimulated the proliferation of N-fixing bacteria. Magnesium decreased the abundance of N-fixing bacteria, whereas liming promoted their growth.

The counts of N-fixing bacteria in soil varied widely from 18 × 10^8^ to 247 × 10^8^ CFU kg^−1^ DM soil (Figure 5c). Average microbial counts were similar in April and August. The abundance of N-fixing bacteria was nearly two-fold higher in May and 21% lower in September relative to May.

##### Actinobacteria

Long-term manure soil application as well as mineral fertilization significantly modified actinobacteria counts in soil (Figure 6a,b). Actinobacteria counts were double the amount higher in soil regularly amended with manure than in soil supplied with mineral fertilizers only (Figure 6a). Lower N rates (30 and 60 kg ha^−1^) did not have a highly stimulating effect on actinobacteria counts. Only the highest N rate (90 kg ha^−1^) induced a 15% increase in the abundance of actinobacteria relative to the control (without mineral fertilization). According to Vetanovetz and Peterson [50], mineral N fertilization increases actinobacteria counts in soil. In the current study, the growth of actinobacteria was stimulated by higher K rates (66.4 and 99.7 kg ha^−1^). The highest K rate induced the greatest (2-fold) increase in actinobacteria counts relative to the lowest K rate. Magnesium did not influence the abundance of the studied bacterial group. Actinobacteria counts clearly increased in regularly limed soil.

Actinobacteria counts varied considerably during the growing season of 2015 (Figure 6c). The mean abundance of actinobacteria increased steadily between April and September. Barabasz and Vořišek [51] and Natywa et al. [52] reported the highest actinobacteria counts in summer, which could be attributed to high temperatures.

##### Fungi

Fungal abundance was 32% higher in soil amended with manure every other year than in soil supplied with mineral fertilizers only (Figure 7a). Fungal counts also increased in response to higher N rates (Figure 7b). Similar observations were made by Natywa et al. [52], Sosnowski et al. [53] and Sosnowski and Jankowski [54]. According to Niewiadomska et al. [55], N fertilization considerably increased fungal abundance relative to control soil. In the work of Wyszkowska [31], increasing urea rates also led to a significant increase in fungal counts in soil. In the present study, fungal abundance was 1.7-fold higher in soil supplied with the highest K rate than in soil fertilized with N_2_P_1_K_1_. Regular liming also enhanced fungal growth in soil.

In the growing season of 2015, fungal counts in soil ranged from 11 × 10^6^ to 250 × 10^6^ CFU kg^−1^ DM soil (Figure 7c). Mean fungal counts were highest in May (130 × 10^6^ CFU kg^−1^ DM soil) and lowest in August. The analyzed parameter was similar in early spring (April) and late summer (September).

#### 3.2.2. Enzymatic Activity

##### Dehydrogenases

Dehydrogenases (DHA) are regarded as reliable indicators of soil biochemical activity. Dehydrogenase activity is influenced by enzymes secreted by soil-dwelling microorganisms, both aerobic and anaerobic [56]. Dehydrogenases determine soil quality and fertility [57]. Ciarkowska and Gambuś [58] reported a strong correlation between DHA activity and organic carbon content in soil. In the present study, manure and mineral fertilization modified DHA activity in soil (Figure 8a,b). Dehydrogenase activity was 1.8-fold higher in soil with manure application than in soil supplied with mineral fertilizers only (Figure 8a). Manure exerted similar effects on DHA activity in the work of Koper and Siwik-Ziomek [59] and Saha et al. [60]. According to Piotrowska and Koper [61] and Natywa et al. [52], DHA activity in soil increased in response to organic amendments and decreased in response to mineral fertilizers (NPK+Ca). In turn, Kucharski and Wałdowska [62] found that mineral fertilizers stimulated DHA activity, but to a smaller extent than organic amendments.

A comparison of the observed changes in DHA activity revealed that the lowest N rate used in the study (30 kg ha^−1^) decreased the analyzed parameter by 8% relative to the control treatment (Figure 8c). However, DHA activity decreased in response to higher N rates (60 and 90 kg N ha^−1^). Kucharski [63], Lemanowicz and Koper [64] and Niewiadomska et al. [55] also found that higher N rates suppressed DHA activity in soil. In contrast, potassium did not inhibit DHA activity and even increased the studied parameter. In a study by Koper and Siwik-Ziomek [59], comprehensive mineral and organic fertilization with calcium and magnesium enhanced the biochemical activity of soil-dwelling microorganisms, increased DHA activity and promoted microbial growth. In the current experiment, regular soil liming enhanced DHA activity by increasing soil pH and reducing hydrolytic acidity. Zaborowska et al. [65] also reported that DHA activity decreased more than three-fold when soil pH was reduced from 7.1 to 6.4. Kalembasa and Kuziemska [66] found that soil liming stimulated DHA activity.

Dehydrogenase activity in soil was determined in the range of 2.13 to 9.65 µmol TFF kg^−1^ DM h^−1^ during the growing season (Figure 8c). This parameter peaked in August 2015 (5.75 µmol TFF kg^−1^ DM h^−1^) and was only somewhat lower in September (5.26 µmol TFF kg^−1^ DM h^−1^). In May, DHA activity was 20% higher than in April. The observed variations in the studied parameter could be attributed to changes in the moisture and oxygen content of soil (Appendix A).

##### Catalase

Catalase is an antioxidant enzyme that protects plants against abiotic and biotic factors that cause oxidative stress [67]. Manure amendment increased catalase activity in soil (Figure 9a). The value of this parameter was 17% higher in the second year after manure application than in soil supplied with mineral fertilizers only. In a study by Lemanowicz and Koper [64], catalase activity also increased in treatments where maize was amended with manure. In the present study, the lowest N rate (30 kg ha^−1^) had no significant effect on catalase activity in soil (Figure 9b). In turn, the highest N rate (90 kg ha^−1^) increased catalase activity. Increasing N rates also stimulated catalase activity in the work of Lemanowicz and Koper [64,68]. Potassium and magnesium fertilizers stimulated catalase activity in soil. Regular liming was particularly effective in enhancing catalase activity and it increased the analyzed parameter 1.4-fold relative to the treatment fertilized with N_2_P_1_K_2_Mg. Catalase activity in soil varied during the growing season of 2015 (Figure 9c). The highest value was noted in September, followed by August; it was the lowest in May.

##### Urease

Regular supply of manure increased organic matter content and stimulated urease activity in soil (Figure 10a). Kucharski et al. [69] also found that manure application significantly enhanced urease activity in tested soils. In our study, urease activity was not significantly modified by mineral fertilization (Figure 10b). However, soil liming exerted a positive effect on urease activity.

In the growing season of 2015, urease activity ranged from 0.02 to 0.46 mmol N-NH_4_ kg^−1^ soil h^−1^ (Figure 10c). The studied parameter was highest in September (0.23 mmol N-NH_4_ kg^−1^ soil h^−1^) and lowest in May (0.04 mmol N-NH_4_ kg^−1^ soil h^−1^). Urease activity was 1.5 times higher in August (0.18 mmol N-NH_4_ kg^−1^ soil h^−1^) than in April.

##### Acid Phosphatase

Acid phosphatase activity differed significantly between treatments treated by manure and treatments supplied with mineral fertilizers only (Figure 11a,b). In soil regularly amended with manure, acid phosphatase activity was 1.7 times higher than in soil supplied with mineral fertilizers only. Lemanowicz and Koper [70] also found that acid phosphatase activity was lower when manure was not applied.

In turn, mineral fertilizers had no significant influence on the activity of the discussed enzyme. However, higher N rates can stimulate acid phosphatase activity by increasing the concentration of H+ in the soil solution as a result of nitrification and enhancing NH4+ uptake by plants. The highest N rate applied (90 kg N ha^−1^) induced the greatest (20%) increase in acid phosphatase activity relative to the control treatment. In the work of Kucharski [63], very high N rates (240 kg ha^−1^) stimulated the activity of acid phosphatase. Lemanowicz and Koper [64,70], Lemanowicz [41] and Siwik-Ziomek and Lemanowicz [47] also reported an increase in acid phosphatase activity with a rise in mineral N rates. The cited authors observed that high N rates stimulated the activity of acid phosphomonoesterase. In contrast, liming induced a minor decrease in the studied parameter. Phosphomonoesterases are highly sensitive to changes in pH and the optimal soil pH for acid phosphatase is 4.0–6.5 [71]. Kuziemska et al. [72] found that soil liming significantly decreased acid phosphatase activity regardless of year or sampling date.

Acid phosphatase activity ranged from 2.94 to 14.66 mmol PN kg^−1^ h^−1^ in the growing season of 2015 (Figure 11c). The analyzed parameter was highest in August and September and much lower in April and May. According to Natywa et al. [52], acid phosphatase activity increases in fall due to the supply of fresh organic matter with harvest residues that stimulate microbial growth. Lemanowicz and Krzyżaniak [73] observed that enzymatic processes are difficult to interpret during the growing season because they are largely influenced by changes in temperature and soil moisture content.

##### Alkaline Phosphatase

Long-term manure amendment and mineral fertilization modified alkaline phosphatase (AlP) activity in soil (Figure 12a,b). In soil amended with manure every other year, this parameter was 2.3 times higher than in treatments supplied with mineral fertilizers only. According to research, organic phosphorus enhances alkaline phosphatase activity in soil [70,74]. Sienkiewicz et al. [75] found that prolonged manure amendment increased the content of available phosphorus in soil. Lemanowicz and Koper [70] reported strong correlations between the content of organic and plant-available phosphorus vs. phosphatase activity. In their opinion, phosphatase activity is indicative of phosphorus levels in soil.

Alkaline phosphatase activity was stimulated by the lowest N rate (30 kg N ha^−1^) and suppressed by higher N rates (60 and 90 kg ha^−1^). In the work of Lemanowicz and Koper [70], an N rate of 90 kg N ha^−1^ also induced a significant (13%) decrease in AlP activity. Higher N rates also inhibited AlP activity in a study by Lemanowicz [41]. Kucharski [63] reported that a very high N rate (240 kg ha^−1^) decreased the value of this parameter in soil. In the current experiment, regular soil liming increased AlP activity two-fold relative to the treatment fertilized with N_2_P_1_K_2_Mg. Similar results were reported by Kalembasa and Kuziemska [66] and Kuziemska et al. [72]. Liming enhances soil enzymatic activity because nutrients are more available in soils with a near-neutral pH [65,66]. Lemanowicz [41] found a correlation between AlP activity and hydrolytic acidity. Similar observations were made in the present study, where AlP activity decreased with a rise in hydrolytic acidity (Table 1).

Alkaline phosphatase activity fluctuated in the growing season of 2015 (Figure 12c). This parameter was highest in April (1.75 mmol PN kg^−1^ h^−1^) and the values noted in May and August were similar. Higher AlP activity in spring could be associated with rapid phosphorus uptake by plant roots and the resulting decrease in the content of available phosphorus in soil. Such conditions support the secretion of phosphatases by plant roots, which catalyze the hydrolysis of organic phosphorus compounds to mineral compounds [73]. According to Lemanowicz and Bartkowiak [76], phosphatase secretion by roots and microorganisms is determined by the plants’ phosphorus requirements. In the present study, alkaline phosphatase activity was lowest in September (1.08 mmol PN kg^−1^ h^−1^).

### 3.3. Content of PAHs in Soil

Statistical analyses revealed that manure (M), mineral fertilization (Min) and M × Min interactions significantly influenced the total content of 16 PAHs and the content of LMW PAHs (naphthalene, acenaphthene, acenaphthylene, fluorene, anthracene, phenanthrene, fluoranthene, pyrene and chrysene) and HMW PAHs (benzo(a)anthracene, benzo(a)pyrene, benzo(b)fluoranthene, benzo(k)fluoranthene, benzo(g,h,i)perylene, indeno(1,2,3-cd)pyrene and dibenzo(a,h)anthracene) (Table 3). In 2015, the total content of PAHs (16) and the content of LMW PAHs was higher in soil amended with manure than in soil supplied with mineral fertilizers only (the effect of manure) (Appendix A). The content of PAHs in soil varied significantly across sampling dates (Appendix A).

#### 3.3.1. Content of LMW PAHs in Soil

The content of LMW PAHs in soil differed significantly during the growing season (Appendix A, Figure 13); it was highest in May (384.7 µg kg^−1^) and lowest in August (119.8 µg kg^−1^). This value was significantly higher in April (259.5 µg kg^−1^) than in September (210.0 µg kg^−1^).

In the growing season of 2015, the content of LMW PAHs (naphthalene, acenaphthene, acenaphthylene, fluorene, anthracene, phenanthrene, fluoranthene, pyrene and chrysene) was highly similar in soil treated with manure and in soil supplied with mineral fertilizers only (Figure 14). The analyzed parameter was higher between April and August in soil treated by manure and in September in treatments supplied with mineral fertilizers. In April and September, the content of LMW PAHs was identical in soil supplied with mineral fertilizers only.

#### 3.3.2. Content of HMW PAHs in Soil

The content of HMW PAHs (benzo(a)anthracene, benzo(a)pyrene, benzo(b)fluoranthene, benzo(k)fluoranthene, benzo(g,h,i)perylene, indeno(1,2,3-cd)pyrene and dibenzo(a,h)anthracene) in soil differed significantly during the growing season (Appendix A, Figure 15). The analyzed parameter was highest in September (158.3 µg kg^−1^) and lowest in August (75.0 µg kg^−1^).

The content of HMW PAHs was lower in April, May and August, and in September in soil supplied with mineral fertilizers only (Figure 16). In April, the greatest difference in the analyzed parameter was observed between soil with manure treatment (81.5 µg kg^−1^) and soil supplied with mineral fertilizers only (117.0 µg kg^−1^).

#### 3.3.3. Total Content of 16 PAHs

The total content of 16 PAHs in soil varied significantly in the growing season of 2015 (Appendix A, Figure 17). The fluctuations in the analyzed parameter could have resulted from varied weather conditions. According to Eriksson et al. [77], low temperatures significantly decrease the rate of PAH degradation in soil. Wang et al. [78] observed that, in periods of heavy rainfall, atmospheric PAHs are transported to soil and tend to accumulate in the soil environment. The total content of PAHs was lowest in August (194.8 µg kg^−1^) and highest in May (484.6 µg kg^−1^). The analyzed parameter was significantly lower in April (358.7 µg kg^−1^) than in September (368.3 µg kg^−1^). According to the IUNG system [79], soil can be classified as non-contaminated (i.e., with ∑13PAH concentrations < 600 µg kg^−1^).

Microbial abundance and soil enzymatic activity undoubtedly influenced the observed fluctuations in the total content of 16 PAHs. The examined parameter was lowest in August when dehydrogenase activity in soil was much higher (Figure 18). In a study by Maliszewska-Kordybach and Smreczak [29], high PAH levels inhibited the activity of dehydrogenases, which is highly sensitive to these pollutants. In the present study, fungal abundance was highest in May (soil most contaminated with PAHs) (Figure 7). Gałązka et al. [80] also reported an increase in fungal counts with a raise in anthracene levels in soil. Samanta et al. [81] emphasized the important role of the biodegradation of PAHs in the soil environment and compared their activity with that of bacteria. In the current study, the total content of 16 PAHs was higher in soil amended with manure on all sampling dates.

### 3.4. Principal Component Analysis: Correlations between Selected Soil Properties

The presence of correlations between selected properties of soil samples collected on four dates in 2015 was identified by principal component analysis (PCA). In April, the first two principal components explained 65% of total variance in the following variables: abundance of organotrophic bacteria, ammonifying bacteria, nitrogen-fixing bacteria, actinobacteria and fungi; activity of dehydrogenases, catalase, urease, acid phosphatase and alkaline phosphatase; content of total nitrogen and organic carbon; Hh and pH; content of LMW PAHs; content of HMW PAHs; total content of 16 PAHs (Figure 19, Appendix A). The analyzed parameters were grouped on one side of the PC1 axis and the total variance explained by these components was very high at 48.3%. Microbial counts (organotrophic bacteria, ammonifying bacteria, N-fixing bacteria and actinobacteria) were strongly correlated with alkaline phosphatase activity in soil. An analysis of the first principal component (PC1) also revealed strong negative correlations between Hh values vs. the activity of catalase, dehydrogenases and urease; pH; total nitrogen content; organic carbon content; total content of 16 PAHs; content of LMW PAHs. An analysis of the second principal component (PC2) demonstrated that the negative correlation between acid phosphatase activity and the content of HMW PAHs explained 16.7% of total variance in the examined soil properties.

The influence of enzymatic activity on the studied soil parameters increased in May. The strong correlations between the activity of dehydrogenases, catalase, urease, acid phosphatase and alkaline phosphatase, pH, organic carbon content, total nitrogen content and actinobacteria counts explained 41.4% of total variance (Figure 20, Appendix A). An analysis of PC2 revealed strong correlations between the total content of 16 PAHs, content of LMW PAHs, counts of organotrophic bacteria and hydrolytic acidity.

In August, PC1 explained 42.1% of total variance in the examined soil parameters. The abundance of organotrophic bacteria and actinobacteria and soil enzymatic activity (dehydrogenases, urease and acid and alkaline phosphatase) were strongly linked with pH and organic carbon and total nitrogen content (Figure 21, Appendix A). Similar to the previous sampling date, the studied parameters were bound by a strong negative correlation with Hh values. An analysis of PC2 revealed that the strong correlation between the counts of nitrogen-fixing bacteria and the content of HMW PAHs explained 17.0% of total variance.

Soil samples collected in September were also characterized by high levels of microbial and enzymatic activity. High microbial abundance can be attributed to a higher content of organic matter that was supplied to soil with harvest residues. An analysis of PC1 demonstrated that strong correlations between all microbial counts (organotrophic, ammonifying, N-fixing bacteria and actinobacteria), enzymatic activity (dehydrogenases, catalase, urease and acid and alkaline phosphatase), pH (in 1 mol KCl) and the content of organic carbon and total nitrogen explained 50.5% of total variance (Figure 22, Appendix A). An analysis of PC2 also revealed that total PAH content and the content of LMW and HMW PAHs in soil were strongly correlated.

Microbial abundance increases under supportive conditions for microbial growth [67]. According to Wielgosz and Szember [82], microbial counts tend to be higher in two periods of the year: in spring, when temperature and soil moisture content increase, and in fall, when fresh organic matter is supplied to the soil environment with harvest residues. Natywa et al. [52] and Wielgosz and Szember [82] also observed that the increase in the abundance of soil-dwelling microorganisms in fall is directly linked with the additional supply of organic matter in the form of harvest residues. Sosnowski et al. [54] reported higher soil microbial counts in fall than in spring, regardless of the experimental factors, and attributed their findings to higher precipitation in fall.

In the work of Lemanowicz and Bartkowiak [76], acid phosphatase activity was highly correlated with the organic carbon content of soil. In the present study, the above correlation was noted in soil samples collected between May and September.

According to Dąbek-Szreniawska et al. [49], soil pH has a considerable influence on enzymatic activity. In the current experiment, hydrolytic acidity had a negative effect on the activity of soil enzymes, excluding acid phosphatase and catalase. Natywa et al. [52] found that dehydrogenase activity was significantly affected by pH and the content of organic carbon and total nitrogen in soil. Ciarkowska and Gambuś [58] also reported a strong correlation between dehydrogenase activity and organic carbon content. In turn, Zaborowska et al. [65] found that dehydrogenase activity was strongly affected by soil pH.

In a study by Maliszewska-Kordybach and Smreczak [83], soil contamination with PAHs inhibited dehydrogenase activity. Lipińska et al. [84] observed that dehydrogenases were more resistant to PAH pollution than urease. According to Wyszkowska and Wyszkowski [85], Lipińska et al. [74] and Lipińska et al. [84], urease activity is compromised in soils heavily contaminated with PAHs. The presence of correlations between LMW PAHs (fluorene, fluoranthene and anthracene) and dehydrogenase activity was also reported by Klimkowicz-Pawlas and Maliszewska-Kordybach [86] and Oleszczuk et al. [18]. The content of PAHs is determined by the concentration of organic carbon and total nitrogen in soil [43,87,88]. In the present study, organic carbon and total nitrogen concentrations were strongly correlated with the total content of PAHs and the content of LMW PAHs in soil samples collected in early spring (Appendix A). Maliszewska-Kordybach et al. [89], Wyszkowski and Ziółkowska [90] and Jin et al. [91] also observed significant correlations between organic carbon content and PAH levels in soil. In contrast, organic carbon content had no significant impact on PAH levels in soil in a study by Bi et al. [92].

Gałązka et al. [80] demonstrated that the content of HMW PAHs was negatively correlated with acid phosphatase activity. The above correlation was also noted in this study in soil samples collected in early spring. In turn, Gałązka et al. [80] found that fungal abundance increased with a rise in anthracene levels in soil. According to Samanta et al. [81], fungi and bacteria play an equally important role in PAH biodegradation in soil. Lehmann et al. [93] demonstrated that an increase in soil organic carbon content stimulated microbial activity and minimized the toxic effects of soil pollutants.

Soil is a complex matrix whose physical, physicochemical, chemical and biological properties are correlated with microbial activity and the presence of pollutants such as PAHs. Weather fluctuations during the growing season also exert a strong influence on chemical and biochemical processes in soil. The relationships between the examined soil parameters were, at least partly, identified in PCA. The PCA revealed that biological processes in soil are determined mainly by carbon and nitrogen content in soil, soil pH and Hh values. Microbial proliferation rates affect soil enzymatic activity. However, the impact of specific microbial groups on PAH levels in soil could not be determined based on the results of a short-term study. Data covering a longer period of time are also needed to formulate reliable conclusions about the impact of PAHs on soil enzymatic activity. However, the present findings indicate that PCA should be used to evaluate the relationships between diverse soil parameters.

## 4. Conclusions

The study revealed considerable seasonal variations in PAH levels in soil, depending on weather conditions and the activity of soil-dwelling microorganisms. The total content of 16 PAHs and the content of LMW and HMW PAHs was higher in soil amended with manure than in soil supplied with mineral fertilizers only. Manure application increased organic carbon and total nitrogen content, stimulated the activity of organotrophic, ammonifying and nitrogen-fixing bacteria, actinobacteria and fungi and increased the activity of dehydrogenases, catalase, urease and acid and alkaline phosphatase. Rising N rates increased the abundance of organotrophic bacteria and fungi and enhanced acid phosphatase activity in soil but inhibited the activity of dehydrogenases and alkaline phosphatase. Soil liming was most effective in increasing the counts of ammonifying bacteria, nitrogen-fixing bacteria, organotrophic bacteria and actinobacteria. Liming also enhanced the activity of catalase, urease and alkaline phosphatase and suppressed acid phosphatase activity. The study showed that manure is one of the important sources of polycyclic aromatic hydrocarbons in the soil. Further research, therefore, is still needed to investigate the effects of applied manure and mineral fertilizers under field conditions on the bioremediation of PAH-polluted soils.

## Figures and Tables

**Figure 1 ijerph-20-03796-f001:**
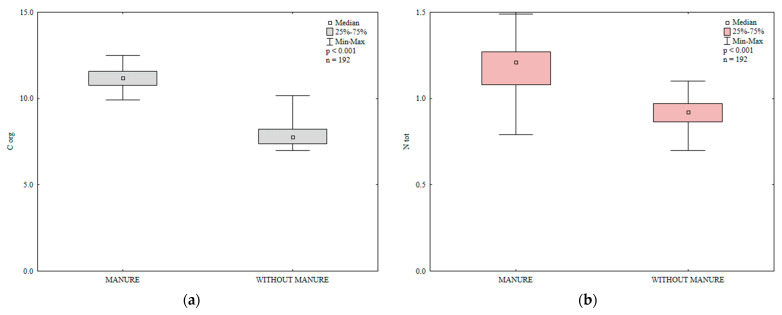
The effect of manure application on the content of organic carbon (Corg (**a**)) and total nitrogen (Ntot (**b**)) in soil (0–30 cm), g kg^−1^.

**Figure 2 ijerph-20-03796-f002:**
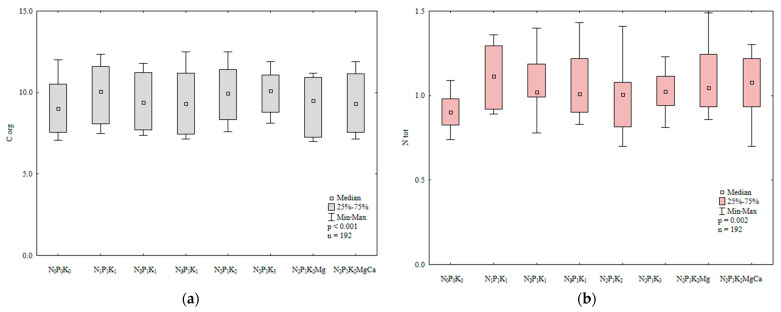
The effect of mineral fertilization on the content of organic carbon (Corg (**a**)) and total nitrogen (Ntot (**b**)) in soil (0–30 cm), g kg^−1^.

**Figure 3 ijerph-20-03796-f003:**
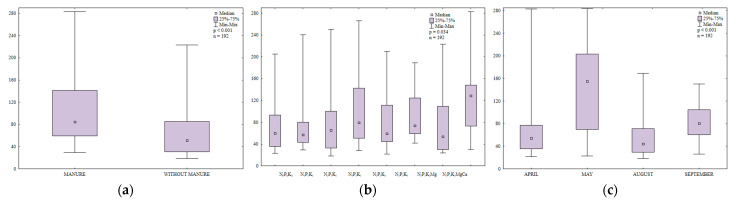
Abundance of organotrophic bacteria in soil amended and not amended with manure (**a**), in soil supplied with mineral fertilizers (**b**) and on different sampling dates (**c**); 10^8^ CFU kg^−1^ DM.

**Figure 4 ijerph-20-03796-f004:**
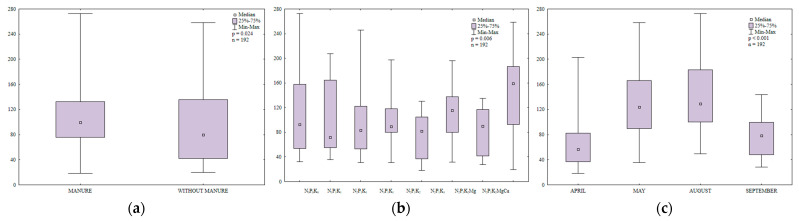
Abundance of ammonifying bacteria in soil amended and not amended with manure (**a**), in soil supplied with mineral fertilizers (**b**) and on different sampling dates (**c**); 10^8^ CFU kg^−1^ DM.

**Figure 5 ijerph-20-03796-f005:**
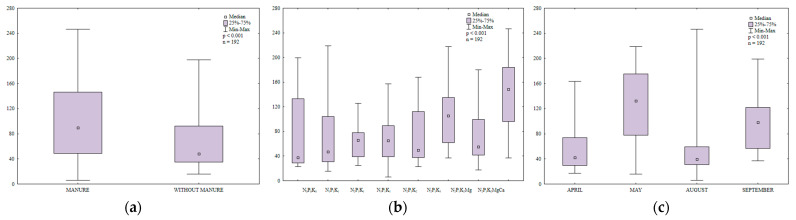
Abundance of nitrogen-fixing bacteria in soil amended and not amended with manure (**a**), in soil supplied with mineral fertilizers (**b**) and on different sampling dates (**c**); 10^8^ CFU kg^−1^ DM.

**Figure 6 ijerph-20-03796-f006:**
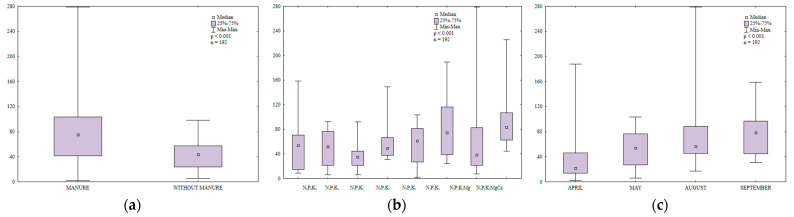
Abundance of actinobacteria in soil amended and not amended with manure (**a**), in soil supplied with mineral fertilizers (**b**) and on different sampling dates (**c**); 10^6^ CFU kg^−1^ DM.

**Figure 7 ijerph-20-03796-f007:**
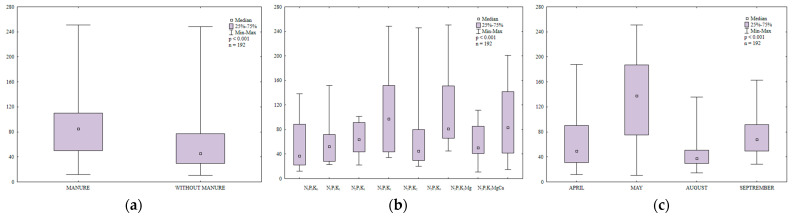
Abundance of fungi in soil amended and not amended with manure (**a**), in soil supplied with mineral fertilizers (**b**) and on different sampling dates (**c**); 10^6^ CFU kg^−1^ DM.

**Figure 8 ijerph-20-03796-f008:**
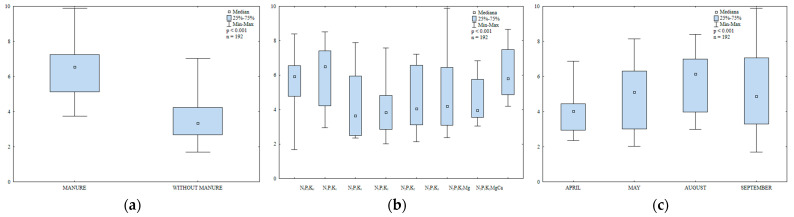
Dehydrogenase activity in soil amended and not amended with manure (**a**), in soil supplied with mineral fertilizers (**b**) and on different sampling dates (**c**); µmol TFF kg^−1^ h^−1^.

**Figure 9 ijerph-20-03796-f009:**
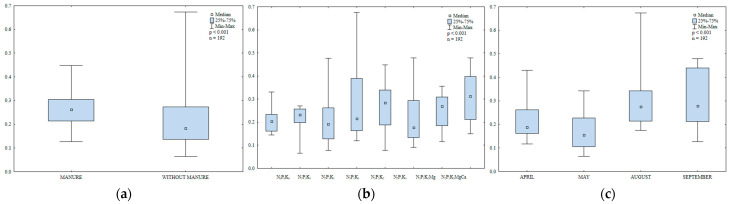
Catalase activity in soil amended and not amended with manure (**a**), in soil supplied with mineral fertilizers (**b**) and on different sampling dates (**c**); mol O_2_ kg^−1^ h^−1^.

**Figure 10 ijerph-20-03796-f010:**
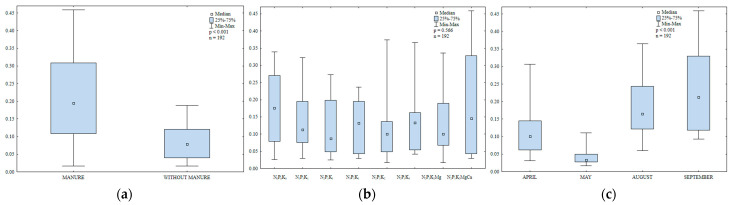
Urease activity in soil amended and not amended with manure (**a**), in soil supplied with mineral fertilizers (**b**) and on different sampling dates (**c**); mmol N-NH_4_ kg^−1^ h^−1^.

**Figure 11 ijerph-20-03796-f011:**
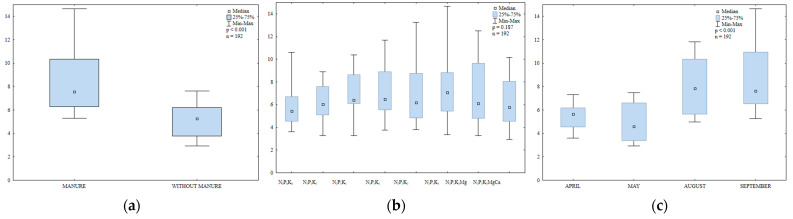
Acid phosphatase activity in soil amended and not amended with manure (**a**), in soil supplied with mineral fertilizers (**b**) and on different sampling dates (**c**); mmol PN kg^−1^ h^−1^.

**Figure 12 ijerph-20-03796-f012:**
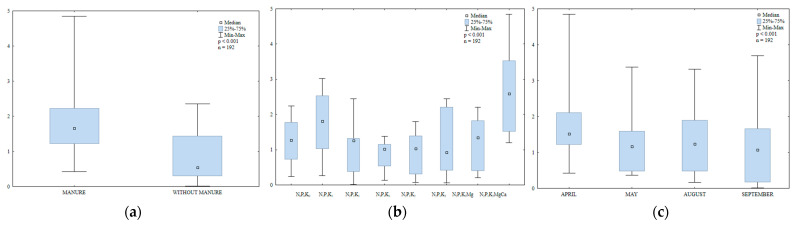
Alkaline phosphatase activity in soil amended and not amended with manure (**a**), in soil supplied with mineral fertilizers (**b**) and on different sampling dates (**c**); mmol PN kg^−1^ h^−1^.

**Figure 13 ijerph-20-03796-f013:**
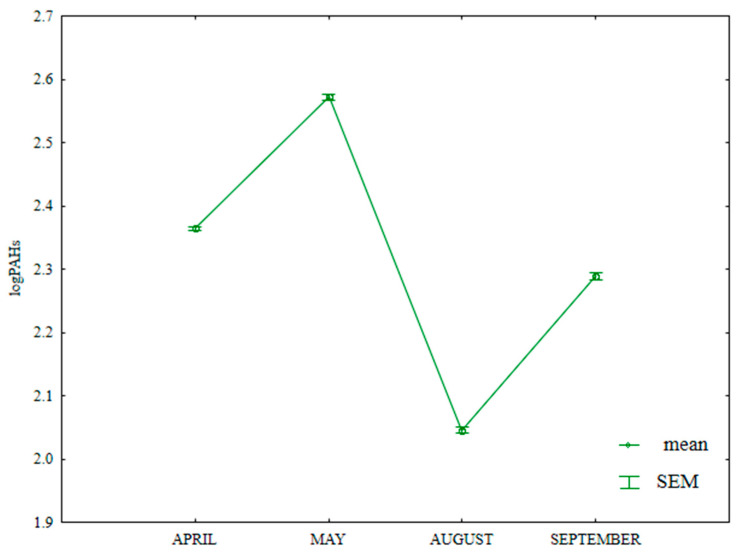
Content of LMW PAHs in soil supplied with mineral fertilizers (log transformed data for 2015). SEM—standard error of the mean.

**Figure 14 ijerph-20-03796-f014:**
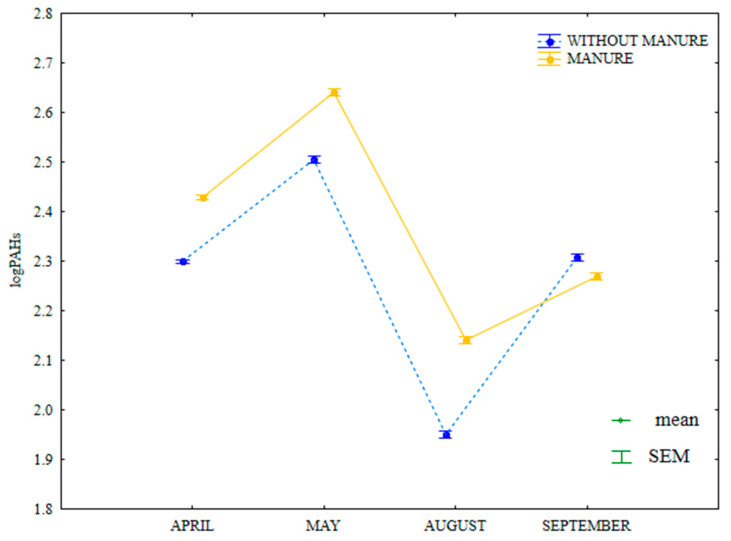
Content of LMW PAHs in soil amended and not amended with manure (log transformed data for 2015). SEM—standard error of the mean.

**Figure 15 ijerph-20-03796-f015:**
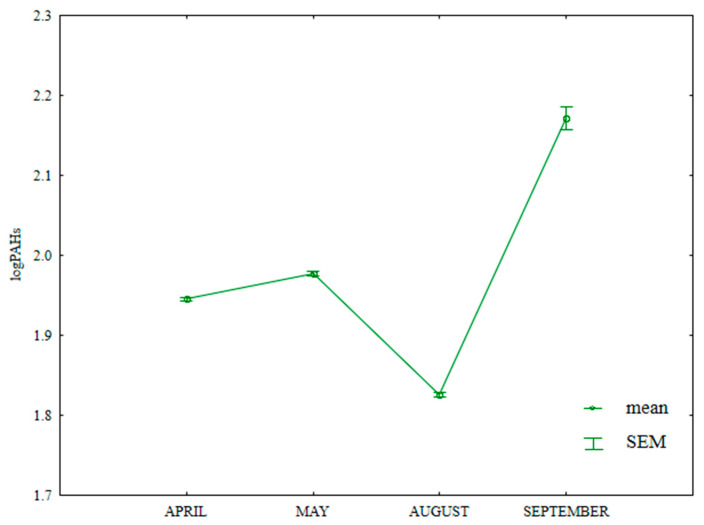
Content of HWM PAHs in soil supplied with mineral fertilizers (log transformed data for 2015). SEM—standard error of the mean.

**Figure 16 ijerph-20-03796-f016:**
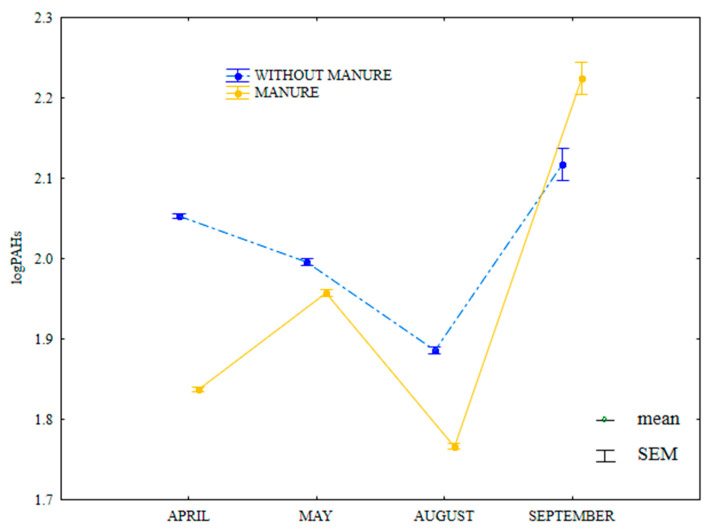
Content of HWM PAHs in soil amended and not amended with manure (log transformed data for 2015). SEM—standard error of the mean.

**Figure 17 ijerph-20-03796-f017:**
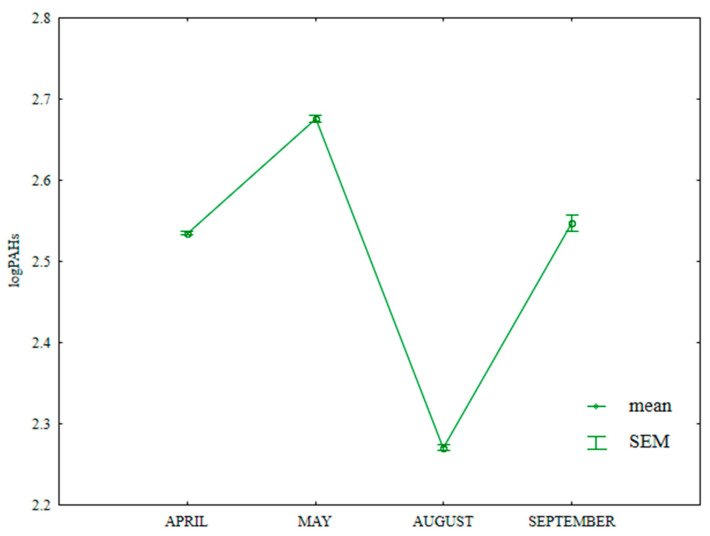
Total content of 16 PAHs in soil supplied with mineral fertilizers (log transformed data for 2015). SEM—standard error of the mean.

**Figure 18 ijerph-20-03796-f018:**
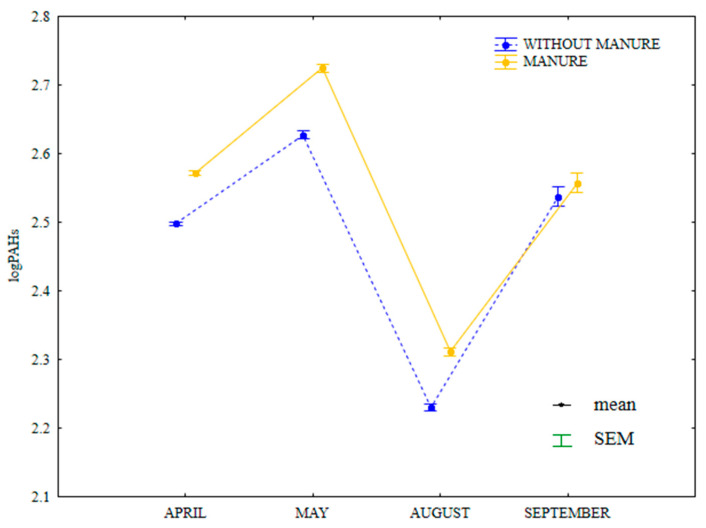
Total content of 16 PAHs in soil amended and not amended with manure (log transformed data for 2015). SEM—standard error of the mean.

**Figure 19 ijerph-20-03796-f019:**
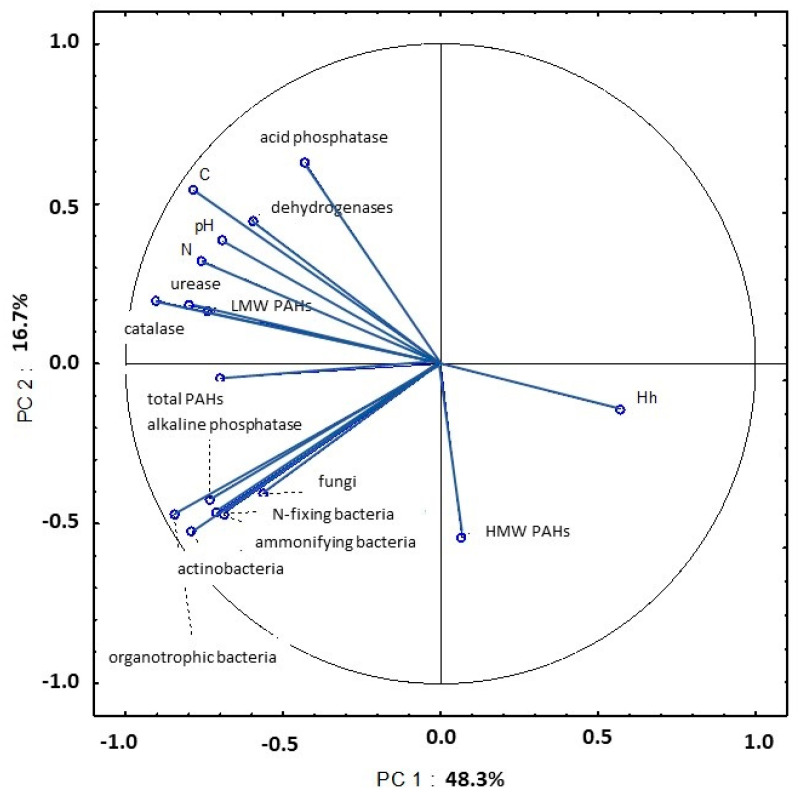
Correlations between the first two principal components and the analyzed variables (April 2015).

**Figure 20 ijerph-20-03796-f020:**
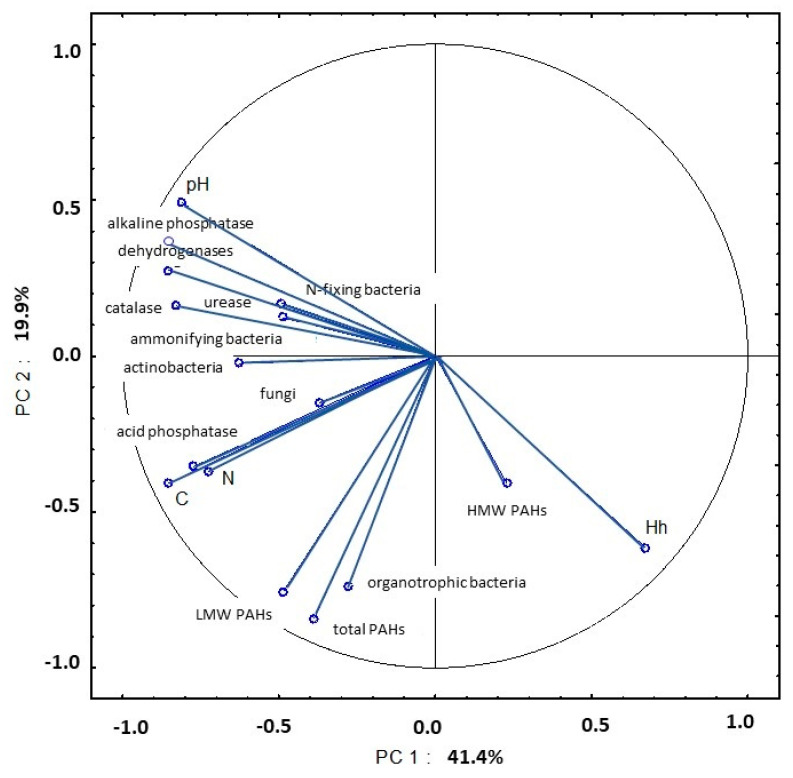
Correlations between the first two principal components and the analyzed variables (May 2015).

**Figure 21 ijerph-20-03796-f021:**
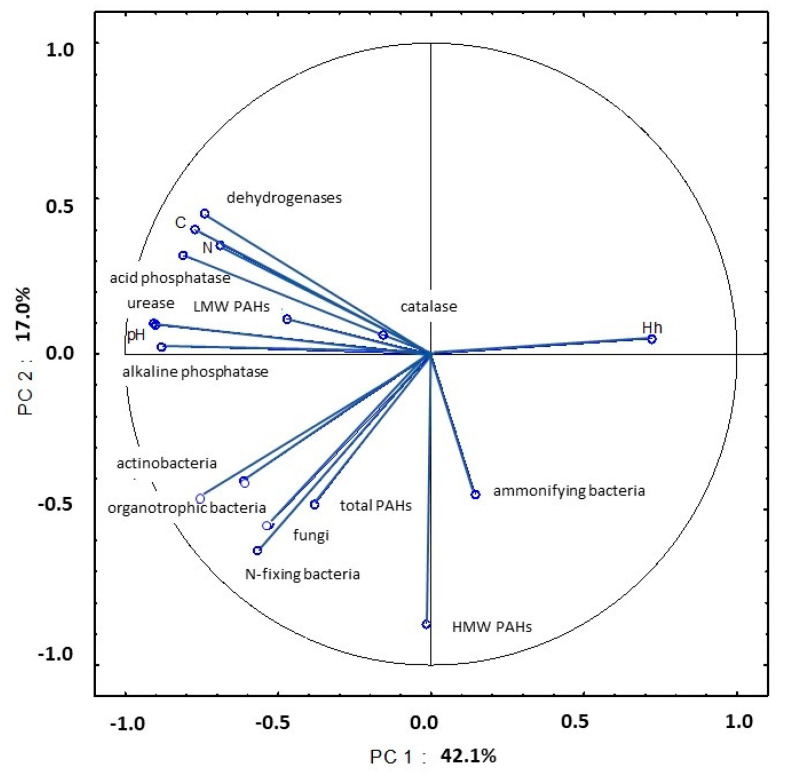
Correlations between the first two principal components and the analyzed variables (August 2015).

**Figure 22 ijerph-20-03796-f022:**
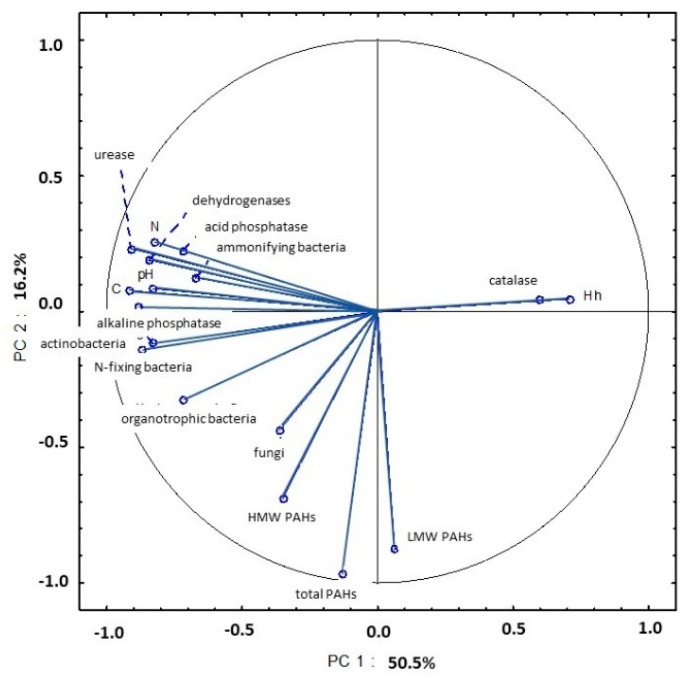
Correlations between the first two principal components and the analyzed variables (September 2015).

**Table 1 ijerph-20-03796-t001:** Soil pH and hydrolytic acidity (0–30 cm).

Treatment	N_0_P_0_K_0_	N_1_P_1_K_1_	N_2_P_1_K_1_	N_3_P_1_K_1_	N_2_P_1_K_2_	N_2_P_1_K_3_	N_2_P_1_K_2_Mg	N_2_P_1_K_2_MgCa
pH (1 mol KCl dm^−3^)								
Manure	5.7	5.8	5.3	5.1	5.2	5.3	5.2	6.3
Without manure	5.0	4.9	4.6	4.5	4.6	4.6	4.6	5.5
Hh (mmol^(+)^ kg^−1^)								
Manure	17.7	16.7	25.6	26.9	25.9	24.2	25.0	13.2
Without manure	25.3	26.3	31.6	34.0	29.6	29.2	27.3	14.0

**Table 2 ijerph-20-03796-t002:** Two-way repeated measures analysis of variance of the organic carbon and total nitrogen content of soil.

Source of Variation	df	Organic Carbon	Total Nitrogen
Manure (M)	1	**	*
Block (B)	2	ns	ns
Mineral fertilization (Min)	7	**	**
M × Min	7	**	**
Error 1	30	-	-
Date (D)	3	**	**
D × M	3	**	**
D × B	6	ns	ns
D × Min	21	**	**
D × M × Min	21	**	**
Error 2	90	-	-

* significance level *p* < 0.05, ** *p* < 0.01, ns—not significant.

**Table 3 ijerph-20-03796-t003:** Two-way repeated measures analysis of variance of the PAH content of soil.

Source of Variation	df	PAHs		
		LMW (9)	HMW (7)	Total (16)
Manure (M)	1	**	**	**
Block (B)	2	ns	ns	ns
Mineral fertilization (Min)	7	**	**	**
M × Min	7	**	**	**
Error 1	30	-	-	-
Date (D)	3	**	**	**
D × M	3	**	**	**
D × B	6	ns	ns	ns
D × Min	21	**	**	**
D × M × Min	21	**	**	**
Error 2	90	-	-	-

** significance level *p* < 0.01; ns—not significant.

## Data Availability

Not applicable.

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
