# Peer review of "The Effect of Spring Barley Fertilization on the Content of Polycyclic Aromatic Hydrocarbons, Microbial Counts and Enzymatic Activity in Soil"

_ijerph, 2023, doi:10.3390/ijerph20053796_

Round 1

Reviewer 1 Report

The manuscript require minor revisions. First of all, the title does not correspond the conclusions obtained; please, correct it. Moreover, commonly known things sound in Abstract and Introduction, e.g. the role and the danger of PAHs. If possible, make them shorter. To improve the soundness of the results in the future, authors should apply methods of metagenomic analysis and profiling (Effect of chicken manure on soil microbial community diversity in poultry keeping areas (T Minkina)  doi: 10.1007/s10653-022-01447-x).

  1. The main question of the manuscript is in revealing the roles of phytoremediation factors (including fertilizers) in soil microbial and enzymatic activity towards pollutants, PAHs, degradation. 2. The topic is relevant in the field, the manuscript includes the long-time experiment with a huge array of data. 3. The authors summarize multy-aspect resutls, and the data obtained and thoroughly cured make a worthy contribution into the soil science. The last ones may be used by other soil scientists and environmental engineers when planning experiments and works with bioremediation methods. 4. As for methodology, authors in the future studies should master metagenomic approaches in determining status and dynamics of microbial communities in the soil or engage mastered metagenomics experts/bioinformatitians into the further studies. As for the manuscript, it is allowed for publication with the methods and methodology presented by authors. 5. Yes 6. References are appropriate 7. Tables and figures are presented with a high quality, and show the results good.

Author Response

Reviewer 1

The manuscript require minor revisions. First of all, the title does not correspond the conclusions obtained; please, correct it. Moreover, commonly known things sound in Abstract and Introduction, e.g. the role and the danger of PAHs. If possible, make them shorter. To improve the soundness of the results in the future, authors should apply methods of metagenomic analysis and profiling (Effect of chicken manure on soil microbial community diversity in poultry keeping areas (T Minkina) doi: 10.1007/s10653-022-01447-x).

  1. The main question of the manuscript is in revealing the roles of phytoremediation factors (including fertilizers) in soil microbial and enzymatic activity towards pollutants, PAHs, degradation. 2. The topic is relevant in the field, the manuscript includes the long-time experiment with a huge array of data. 3. The authors summarize multy-aspect resutls, and the data obtained and thoroughly cured make a worthy contribution into the soil science. The last ones may be used by other soil scientists and environmental engineers when planning experiments and works with bioremediation methods. 4. As for methodology, authors in the future studies should master metagenomic approaches in determining status and dynamics of microbial communities in the soil or engage mastered metagenomics experts/bioinformatitians into the further studies. As for the manuscript, it is allowed for publication with the methods and methodology presented by authors. 5. Yes 6. References are appropriate 7. Tables and figures are presented with a high quality, and show the results good.
  2. As for methodology, authors in the future studies should master metagenomic approaches in determining status and dynamics of microbial communities in the soil or engage mastered metagenomics experts/bioinformatitians into the further studies. As for the manuscript, it is allowed for publication with the methods and methodology presented by authors.

Response: Thank you for your kind comments. Mastering the metagenomic approach in determining the status and dynamics of soil microbial communities could be helpful in interpreting the results in future studies.

  1. First of all, the title does not correspond the conclusions obtained; please, correct it.

Response: We added this sentence to the Conclusions:  “The study showed that manure is one of the important sources of polycyclic aromatic hydrocarbons in the soil. Further research, therefore, still needed to investigate the effects of applied manure and mineral fertilizers under field conditions on bioremediation of PAHs polluted soils”.

  1. Moreover, commonly known things sound in Abstract and Introduction, e.g. the role and the danger of PAHs. If possible, make them shorter.

Response: It was corrected in the Introduction and Abstract.

Thank you for your very insightful and comprehensive review.

Reviewer 2 Report

Manuscript ID ijerph-2170128 entitled "The effect of spring barley fertilization on the content of polycyclic aromatic hydrocarbons, microbial counts and enzymatic activity in soil" presents the evaluation of the impact of different fertilization on the microbial activity and biochemical properties of soil and the accumulation of PAHs in soil. In the current state, there are several weaknesses and obscure points in the manuscript that the authors should further reconsider and clarify. Comments are listed below:

1.The abstract should contain with some data, concerning the results.

2.In the "Introduction" part, there is lack of information regarding the novelty of the experiment. A high-quality paper has to provide a proper state-of-the-art analysis after the literature review which should correspond to the paper goals. Furthermore, the authors laconically present information about microbial degradation of PAHs and the need for this study. Moreover, at the end of the introduction, the main assumptions of the study and hypotheses have to be properly formulated. 

3.The section of “Method” is not detailed enough especially part of "Microbiological and biochemical analyses of soil". The authors should rewrite of the subsections describing the exact parameters of the tests performed. 

4.Moreover, there is no strict quality assurance and quality control and clean up step for the determination of PAHs in soil samples. As we know, too many interferences exist in these matrixes, especially soil containing high contents of organic matter. The authors should to remove the interferences before instrumental analysis. In addition, please add the instrumental conditions for PAH measurement. Please give more description in the main text or provide the detailed results of PAHs in certified Standard Reference Materials in the Supporting Information.

5.Please re-check the conclusion. Moreover, the authors’ final remarks should include details of any limitations of this study and recommendations for future perspectives.

Author Response

Reviewer 2

Manuscript ID ijerph-2170128 entitled "The effect of spring barley fertilization on the content of polycyclic aromatic hydrocarbons, microbial counts and enzymatic activity in soil" presents the evaluation of the impact of different fertilization on the microbial activity and biochemical properties of soil and the accumulation of PAHs in soil. In the current state, there are several weaknesses and obscure points in the manuscript that the authors should further reconsider and clarify. Comments are listed below:

Response:

  1. The abstract should contain with some data, concerning the results.

Response: The results were added to the abstract.

  1. In the "Introduction" part, there is lack of information regarding the novelty of the experiment. A high-quality paper has to provide a proper state-of-the-art analysis after the literature review which should correspond to the paper goals. Furthermore, the authors laconically present information about microbial degradation of PAHs and the need for this study. Moreover, at the end of the introduction, the main assumptions of the study and hypotheses have to be properly formulated.

Response: It was corrected in the text.

  1. The section of “Method” is not detailed enough especially part of "Microbiological and biochemical analyses of soil". The authors should rewrite of the subsections describing the exact parameters of the tests performed.

Response: It was corrected in the text.

  1. Moreover, there is no strict quality assurance and quality control and clean up step for the determination of PAHs in soil samples. As we know, too many interferences exist in these matrixes, especially soil containing high contents of organic matter. The authors should to remove the interferences before instrumental analysis. In addition, please add the instrumental conditions for PAH measurement. Please give more description in the main text or provide the detailed results of PAHs in certified Standard Reference Materials in the Supporting Information.

Response: In manuscript: “The content of nutrients, heavy metals, and PAHs (LMW and HMW) in manure was described previously by Krzebietke et al. [3]” 142-144

The methods were described in Krzebietke et al. as follows:

The content of heavy and light molecular weight polycyclic aromatic hydrocarbons was determined on a gas chromatograph – mass spectrometer Trace GC Ultra ITQ900 coupled with an autosampler TRIPlus (Fisher Scientific) manufactured by THERMO, and equipped with an FID detector. An analysis of the content of polycyclic aromatic hydrocarbons (PAHs) was accomplished after one-hour extraction of 20 g of soil with 20 cm3 of acetonitrile, using an ultrasound washer and horizontal shaker. The extract thus obtained (10 cm3) was decanted and preliminarily purified on an MPW-350R centrifuge and a solid phase extract SPE station. SPE-NH2/C18 cartridges with the adsorbent weight of 1500 mg and the capacity of 6 cm3 were used. 10 cm3 of methanol was applied to flush the PAHs from the adsorbent, after which the extract was concentrated in a neutral gas (nitrogen) atmosphere up to the volume of 0.2 cm3. The samples prepared as described above were subjected to determinations of PAHs with the GC technique, using an FID detector mounted on an Rxi-5ms column 30 m in length, and the inner diameter of 0.25 mm, where the walls were coated with a 0.25-µm thick layer of liquid stationary phase (SCOT column technology). He at a constant flow rate (3 cm3∙min−1), and H2, air and N2 at the respective flow rates of 35, 350 and 30 cm3∙min−1) served as carrier gases. The temperature regime was as follows: 0–100 °C – 0.2 min; 50 °C∙min−1 – 143 °C – 1.5 min; 8 °C∙min−1 – 180 °C – 0.4 min; 100 °C∙min−1 – 210 °C – 1.5 min; 10 °C∙min−1 – 300 °C – 5 min = 23.39 min. The temperature of detectors was set at 340 °C, while the temperature of the splitless injector was set at 250 °C. Determinations were made based on the reference solution by Restek Corporation, containing a mix of 16 PAHs (naphthalene, acenaphtylene, fluorene, anthracene, phenathrene, fluoranthene, pyrene, chrysene, and a sum of heavy PAHs: benzo(a)anthracene, benzo(a)pyrene, benzo(b)fluoranthene, benzo(k)fluoranthene, benzo(g,h,i)perylene, indeno(1,2,3-cd)pyrene, dibenzo(a,h)anthracene in a concentration of 2000 µg∙cm−3 of each component compound. Working solutions equalled 5, 10, 20, 50, 80, 120 µg∙cm−3 of each of the components. The recovery of PAHs from soil ranged from 84% to 93%, and was considered separately for each of the compounds analysed.”.

We decided to cite only the reference which is available Open Access because this information is very long and it will be as repetition from our previous article.

  1. Please re-check the conclusion. Moreover, the authors’ final remarks should include details of any limitations of this study and recommendations for future perspectives.

Response: It was corrected in the text.

Thank you for your very insightful and comprehensive review.

Reviewer 3 Report

Dear Authors,

Thanks for your valuable research.

1.       Please check the text for grammar, spelling, and punctuation errors carefully.

2.       Please write the abstract as formatted abstract.

3.       Please write the several methods used for PAH bioremediation in soil briefly.

4.        What is the novelty of this study?

5.       Please write the limitations of this study.

6.       Please more describe about PAHs in environment and human exposure.

7.       What is environmental levels of PAHs in soil?

8.       Why chosen these 16 PAHs for your study?

9.       Please cite to the analysis methods and more descried about them.

10.   What is initial and final characterization of soil?

11.   How determine the content of PAHs in the soil?

12.   As some of PAHs are volatile, did you consider this volatile part for removal of PAHs?

13.   Why selected these four dates for sampling? It was better to sampling in four season.

14.   Please improve the resolution of figures.

15.   What soil deep was selected for sampling?

16.   What is optimal C:N:P ratio in the soils?

17.    Please explain numerically about C:N:P content in the text.

18.   Please extent the discussion in each section.

19.   Please describe the mechanism of PAHs degradation in separated section.

20.   Please mention the important information as numerically in the abstract and conclusion.

Author Response

Reviewer 3

Response:

  1. Please check the text for grammar, spelling, and punctuation errors carefully.

Response: The text was checked for grammar and punctuation errors.

  1. Please write the abstract as formatted abstract.

Response: The abstract was formatted according to Guidelines for Authors.

  1. Please write the several methods used for PAH bioremediation in soil briefly.

Response: It was corrected in the article.

  1. What is the novelty of this study?

Response:

In manuscript in line 108 – 109 was sentence “The presence and accumulation of PAHs in soil has not been extensively studied to date, and further research is needed to address this problem”.

Added sentence in line 118 - 124: The combined application of manure and mineral fertilizers has been studied in only a very few research experiments, hence their effect on PAH content in soil is still unexplored. The new insights contribute to a better understanding of PAH biodegradation processes under complex natural conditions. It was assumed that the optimal fertilization both with manure and with mineral fertilizers customized strictly to nutritional requirements to field crops does not exceed the permissible concentrations of the assessed PAHs in the soil.

  1. Please write the limitations of this study.

Response: We added sentence: “The study demonstrated that weather conditions and microbial activity induced considerable seasonal variations in PAHs content.”

  1. What is environmental levels of PAHs in soil?

Response: We added sentence: „According to the IUNG system [79] soil can be classified as non-contaminated (i.e. with ∑13PAH concentrations < 600 µg kg-1).

According to the IUNG system, 10% of soils can be considered as contaminated (∑13 PAH>1000 µg∙kg-1), 14% corresponds to weakly contaminated class (∑13PAH in the limits of 600–1000 µg∙kg-1), and 76% of soils can be classified as non-contaminated, i.e. with ∑13PAH concentrations < 600 µg∙kg-1 (including 22% with PAH content below the background level of 200 µg∙kg-1)” Maliszewska-Kordybach, B., Smreczak, B., Klimkowicz-Pawlas, A., & Terelak, H. (2008). Monitoring of the total content of polycyclic aromatic hydrocarbons (PAHs) in arable soils in Poland. Chemosphere, 73(8), 1284–1291. doi:10.1016/j.chemosphere.2008.07.009

  1. Why chosen these 16 PAHs for your study?

Response: We added sentence: “All samples were analyzed for the 16 PAH priority pollutant listed by US EPA [28].

“In 1984 the United States Environmental Protection Agency (USEPA) designated 16 PAHs as compounds of interest under a suggested procedure for reporting test measurement results (naphthalene, acenaphthylene, acenaphthene, fluorine, phenan-threne, anthracene, fluoranthene, pyrene, benzo(a)anthracene, chrysene, benzo(b)fluoranthene, benzo(k)fluoranthene, benzo(a)pyrene, dibenzo(a,h)anthracene, benzo(g,h,i)perylene, and indeno(1,2,3-c,d)pyrene) as most priority ones to be analyzed in various environmental matrices.” USEPA (U.S. Environmental Protection Agency). Review and evaluation of the evidence for cancer associated with air pollution. EPA-540/5-83-006R. USEPA, Arlington; 1984. yosemite.epa.gov/ee/epalib/eelib.nsf/73bc8d7fb6d3644385256a290076d16f/19ca908d6a92693185256ad20049c9b3!OpenDocument.

  1. Please cite to the analysis methods and more descried about them.

Response: To the paragraph 2.3. we added sentence: “Microorganisms were isolated with the serial dilution method following the procedure described in the study by Wyszkowska i in. [37]. The procedure for the determination of soil enzymatic activity was presented in the study by Borowik i in. [38] and microbial counts. The culture conditions and the exact procedure for the isolation of microorganisms were described in our earlier paper in the study by Borowik i in.[39].

In manuscript: “The content of nutrients, heavy metals, and PAHs (LMW and HMW) in manure was described previously by Krzebietke et al. [3]” 142-144.

The methods were described in Krzebietke et al. as follows:

The content of heavy and light molecular weight polycyclic aromatic hydrocarbons was determined on a gas chromatograph – mass spectrometer Trace GC Ultra ITQ900 coupled with an autosampler TRIPlus (Fisher Scientific) manufactured by THERMO, and equipped with an FID detector. An analysis of the content of polycyclic aromatic hydrocarbons (PAHs) was accomplished after one-hour extraction of 20 g of soil with 20 cm3 of acetonitrile, using an ultrasound washer and horizontal shaker. The extract thus obtained (10 cm3) was decanted and preliminarily purified on an MPW-350R centrifuge and a solid phase extract SPE station. SPE-NH2/C18 cartridges with the adsorbent weight of 1500 mg and the capacity of 6 cm3 were used. 10 cm3 of methanol was applied to flush the PAHs from the adsorbent, after which the extract was concentrated in a neutral gas (nitrogen) atmosphere up to the volume of 0.2 cm3. The samples prepared as described above were subjected to determinations of PAHs with the GC technique, using an FID detector mounted on an Rxi-5ms column 30 m in length, and the inner diameter of 0.25 mm, where the walls were coated with a 0.25-µm thick layer of liquid stationary phase (SCOT column technology). He at a constant flow rate (3 cm3∙min−1), and H2, air and N2 at the respective flow rates of 35, 350 and 30 cm3∙min−1) served as carrier gases. The temperature regime was as follows: 0–100 °C – 0.2 min; 50 °C∙min−1 – 143 °C – 1.5 min; 8 °C∙min−1 – 180 °C – 0.4 min; 100 °C∙min−1 – 210 °C – 1.5 min; 10 °C∙min−1 – 300 °C – 5 min = 23.39 min. The temperature of detectors was set at 340 °C, while the temperature of the splitless injector was set at 250 °C. Determinations were made based on the reference solution by Restek Corporation, containing a mix of 16 PAHs (naphthalene, acenaphtylene, fluorene, anthracene, phenathrene, fluoranthene, pyrene, chrysene, and a sum of heavy PAHs: benzo(a)anthracene, benzo(a)pyrene, benzo(b)fluoranthene, benzo(k)fluoranthene, benzo(g,h,i)perylene, indeno(1,2,3-cd)pyrene, dibenzo(a,h)anthracene in a concentration of 2000 µg∙cm−3 of each component compound. Working solutions equalled 5, 10, 20, 50, 80, 120 µg∙cm−3 of each of the components. The recovery of PAHs from soil ranged from 84% to 93%, and was considered separately for each of the compounds analysed.”.

We decided to cite only the reference which is available Open Access because this information is very long and it will be as repetition from our previous article.

  1. What is initial and final characterization of soil?

Response: It was presented in line 139 -141 and in supplementary material.

  1. How determine the content of PAHs in the soil?

Response: In manuscript “The content of 16 PAHs was determined with the Trace GC/MS Ultra ITQ900 system with a TRIPlus autosampler (Thermo Fisher Scientific) and a flame ionization detector. The total content of 16 PAHs (naphthalene, acenaphthene, acenaphthylene, fluorene, phe-nanthrene, anthracene, fluoranthene, pyrene, benzo(a)anthracene, chrysene, ben-zo(b)fluoranthene, benzo(k)fluoranthene, benzo(a)pyrene, indeno(1,2,3-cd)pyrene, diben-zo(a,h)anthracene, and benzo(g,h,i)perylene) was determined by the method described by Krzebietke et al. [3].”164-170.

  1. As some of PAHs are volatile, did you consider this volatile part for removal of PAHs?

Response: The place of the experiment is far away from highways and large urban agglomerations and industrial centers. These sources are indicated in the article - Krzebietke, S.J.; Mackiewicz-Walec, E.; Sienkiewicz, S.; Wierzbowska, J.; Załuski, D.; Borowik, A. Polycyclic Aromatic Hydrocarbons in Soil at Different Depths under a Long-Term Experiment Depending on Fertilization. Int. J. Environ. Res. Public. Health 2022, 19, 10460, doi:10.3390/ijerph191610460

  1. Why selected these four dates for sampling? It was better to sampling in four season.

Response: We wanted to capture changes in microbial abundance and enzymatic activity over the entire of growth period of spring barley.

  1. Please improve the resolution of figures?

Response: The programming used does not allow to increase the resolution of the drawings.

  1. What soil deep was selected for sampling?

Response: In our manuscript was sentence in line 146-147: “Soil samples for analyses of chemical, biochemical, and microbiological properties and PAHs levels were collected at a depth of 0-30 cm on four dates during the growing season of spring barley (BBCH-10, BBCH-23), after harvest and after skimming.”

  1. What is optimal C:N:P ratio in the soils?

Response: Carbon: Nitrogen: Phosphorus ratio (C:N:P) as recommended by literature for bioremediation (1:0.1:0.01). The pH in these soils facilitates phosphorus availability, one reason why it is easily detectable and abundant.

Kopytko M, Ibarra-Mojica D (2009) Evaluación del potencialde biodegradación de hidrocarburos torales del petróleo(TPH) en suelos contaminados procedentes de Petrosan-tander (Colombia) Inc. Rev Cient La Univ Pontif Boliv3:35–46. https://doi.org/10.18566/puente.v3n1.a04

In our manuscript we don't cite this ratio because that is not aim of research.

  1. Please explain numerically about C:N:P content in the text.

Response: We based our study on the C:N ratio. According to AMEZCUA-ALLIERI et al. (2012), the C:N ratio is an indicator of PAH removal from soil.

Amezcua-Allieri, M.A.; Ávila-Chávez, M.A.; Trejo, A.; Meléndez-Estrada, J. Removal of Polycyclic Aromatic Hydrocar-bons from Soil: A Comparison between Bioremoval and Supercritical Fluids Extraction. Chemosphere 2012, 86, 985–993, doi:10.1016/j.chemosphere.2011.11.032.

  1. Please more describe about PAHs in environment and human exposure.
  2. Please extent the discussion in each section.
  3. Please describe the mechanism of PAHs degradation in separated section.
  4. Please mention the important information as numerically in the abstract and conclusion.

Response: Question 17, 18, 19, 20 – The aim of the research was to determine the effect of cultivation and fertilization of spring barley before and after cultivation and during vegetation on the accumulation of PAHs in the topsoil. The extending of present already enormous work with this information would be a major departure from the research undertaken. In the already published articles from the same experiment, various issues have already been addressed. In order to avoid plagiarism the authors decided to cite these studies in the text of current research. Some of these issues may be used to develop new research and publications.

Thank you for your very insightful and comprehensive review.

Round 2

Reviewer 3 Report

Please use the following related articles in your paper, if possible:

1.      Mosallaei S, Hashemi H, Hoseini M, Dehghani M, Naz A. Polycyclic Aromatic Hydrocarbons (PAHs) in household dust: The association between PAHs, Cancer Risk and Sick Building Syndrome. Building and Environment 229 (2023) 109966. https://doi.org/10.1016/j.buildenv.2022.109966

12.   Mansooreh Dehghani, Simin Nasseri, Hassan Hashemi: Study of the Bioremediation of Atrazine under Variable Carbon and Nitrogen Sources by Mixed Bacterial Consortium Isolated from Corn Field Soil in Fars Province of Iran. Journal of Environmental and Public Health 03/2013; 2013:973165. DOI:10.1155/2013/973165. Pubmed.